# Targeted protein degradation systems to enhance Wnt signaling

**Parthasarathy Sampathkumar[†], Heekyung Jung[†], Hui Chen, Zhengjian Zhang, Nicholas Suen, Yiran Yang, Zhong Huang, Tom Lopez, Robert Benisch, Sung-Jin Lee, Jay Ye, Wen-Chen Yeh, Yang Li\***

Surrozen, Inc, South San Francisco, United States

**\*For correspondence:**
yang@surrozen.com

[†]These authors contributed equally to this work

**Abstract** Molecules that facilitate targeted protein degradation (TPD) offer great promise as novel therapeutics. The human hepatic lectin asialoglycoprotein receptor (ASGR) is selectively expressed on hepatocytes. We have previously engineered an anti-ASGR1 antibody-mutant RSPO2 (RSPO2RA) fusion protein (called SWEETS) to drive tissue-specific degradation of ZNRF3/RNF43 E3 ubiquitin ligases, which achieved hepatocyte-specific enhanced Wnt signaling, proliferation, and restored liver function in mouse models, and an antibody–RSPO2RA fusion molecule is currently in human clinical trials. In the current study, we identified two new ASGR1- and ASGR1/2-specific antibodies, 8M24 and 8G8. High-resolution crystal structures of ASGR1:8M24 and ASGR2:8G8 complexes revealed that these antibodies bind to distinct epitopes on opposing sides of ASGR, away from the substrate-binding site. Both antibodies enhanced Wnt activity when assembled as SWEETS molecules with RSPO2RA through specific effects sequestering E3 ligases. In addition, 8M24-RSPO2RA and 8G8-RSPO2RA efficiently downregulate ASGR1 through TPD mechanisms. These results demonstrate the possibility of combining different therapeutic effects and degradation mechanisms in a single molecule.

## eLife assessment

The manuscript describes a **valuable** method to boost WNT signaling in a tissue-specific manner. The work extends previous data from the authors based on fusing an RSPO2 mutant protein to an antibody that binds ASGR1/2. In the current manuscript, two new antibodies with similar effects are described, that expand this **solid** approach and provide alternatives for potential future clinical applications. This manuscript will be of interest to all scientists studying protein engineering and cellular targeting.

## Introduction

The Wnt ('Wingless-related integration site' or 'Wingless and Int-1' or 'Wingless-Int') signaling pathway is essential for the biogenesis, homeostasis, and regeneration of many organs and tissues (*Steinhart and Angers, 2018*). In the Wnt/β-catenin signaling system, a Wnt ligand simultaneously engages the frizzled (FZD) family of receptors (FZD$_{1-10}$) and co-receptors LRP5 or LRP6 (low-density lipoprotein receptor-related protein 5/6) resulting in nuclear stabilization of β-catenin, which leads to expression of specific target genes responsible for cellular proliferation and regeneration. Thus, targeted delivery of molecules that modulate or stimulate Wnt signaling in a tissue- or cell-specific manner is a promising therapeutic strategy for the treatment of various diseases where tissue regeneration confers therapeutic benefits (*Fowler et al., 2021*; *Liu et al., 2022*; *Nguyen et al., 2022*; *Nusse and Clevers, 2017*; *Xie et al., 2022*).

Wnt signaling is finetuned by a variety of biochemical mechanisms (*Rim et al., 2022*). One regulatory negative-feedback mechanism involves membrane E3 ubiquitin-protein ligases (E3 ligases), ZNRF3 (zinc and ring finger 3), and RNF43 (ring finger protein 43) proteins, which mark FZD and LRP receptors for proteasomal degradation via ubiquitination and endocytosis (*Koo et al., 2012*; *Hao et al., 2012*). In contrast, RSPO proteins (R-spondins 1–4) enhance Wnt signaling by facilitating clearance of the negative regulators ZNRF3/RNF43 through LGR4–6 (leucine-rich repeat-containing G-protein-coupled receptors 4, 5, 6) from the cell membrane via endocytosis of the RSPO:ZNRF3/RNF43:LGR ternary complex (*Carmon et al., 2011*; *Chen et al., 2013*). RSPO proteins bind to ZNRF3/RNF43 and LGR via their Furin Fu1 and Fu2 domains, respectively. Removal of ZNRF3/RNF43 E3 ligases from the cell membrane stabilizes FZD and LRP receptors and amplifies Wnt signaling. Therefore, targeted delivery of RSPOs could have potential therapeutic benefits for disease conditions where Wnt ligands are present but signaling is limited.

ASGR (asialoglycoprotein receptor, also known as ASGPR) mediates clearance of desialylated, galactose- or *N*-acetylgalactosamine-terminating plasma glycoproteins via receptor mediated endocytosis for lysosomal degradation (*Schwartz et al., 1986*; *Weigel and Yik, 2002*). ASGR is thought to exist as a hetero-oligomer of ASGR1 and ASGR2 polypeptides – referred to as H1 and H2, respectively (*Henis et al., 1990*; *Saxena et al., 2002*). Both ASGR1 (H1) and ASGR2 (H2) polypeptides are type-II membrane proteins with a short N-terminal cytosolic domain followed by a single-transmembrane helix and an extracellular region comprising a helical-stalk region that mediate oligomerization via a coiled-coil structure and a carbohydrate recognition domain (CRD) at their C-terminus (*Meier et al., 2000*). Human ASGR1 and ASGR2 share 54% sequence identity. ASGR is a calcium-dependent C-type lectin that is highly expressed in mammalian hepatocytes, and it has been explored for targeted delivery of drugs to the liver (*Huang et al., 2017*; *Sanhueza et al., 2017*).

Taking advantage of the liver-specific ASGR expression and its ability to induce endocytosis and lysosomal degradation of bound proteins, we have previously described a tissue-targeted RSPO mimetic system (termed SWEETS, for **S**urrozen **W**nt signal **E**nhancer **E**ngineered for **T**issue **S**pecificity), where a mutant-RSPO2 domain (RSPO2RA) fused to an anti-ASGR1 antibody, which specifically upregulates Wnt-target genes in hepatocytes, stimulates hepatocyte proliferation, restoring liver function in a diseased-mouse model (*Zhang et al., 2020*). Since RSPO2RA retains the ability to bind the two E3 ligases but loses the ability to bind LGRs, SWEETS-induced proximity of E3 ligases to ASGR1 results in hepatocyte-specific endocytosis and degradation of E3 ligases, leading to stabilization of FZD and enhancement of Wnt signaling.

As a continuation of these efforts, we herein describe the discovery and characterization of two new ASGR antibodies, namely 8M24 and 8G8. High-resolution crystal structures of ASGR1CRD:8M24 and ASGR2CRD:8G8 complexes revealed that the binders have distinct epitopes on opposing surface of ASGR and provided insights into their specificities. SWEETS molecules assembled by fusion of RSPO2RA to 8M24 (8M24-RSPO2RA) and to 8G8 (8G8-RSPO2RA) showed robust Wnt signal activation in cell-based assays. Further analysis also showed that such bispecific molecules not only induce internalization and lysosomal degradation of E3 ligases, they also induce ASGR degradation through proteasomal and lysosomal pathways. These ASGR-targeted SWEETS molecules represent a unique targeted protein degradation (TPD) platform that functions via multiple mechanisms and presents novel opportunities to treat liver diseases using regenerative therapeutics.

## Results

### 8M24 is a human-ASGR1-specific binder and 8G8 binds to both ASGR1 and ASGR2

Mice immunizations were performed using the recombinantly expressed and purified extracellular domains of human ASGR1 (hASGR1, residues 62–291) and human ASGR2 (hASGR2, residues 66–292). Hybridoma lines were screened by ELISA against immunogens and two ASGR CRDs, hASGR1CRD (residues 154–291) and hASGR2CRD (residues 177–311). The selected hybridoma lines 8M24 and 8G8 were sequenced, and recombinant Fabs or IgGs were cloned, expressed, and purified for further characterization. Analytical size-exclusion chromatography (SEC) was used to investigate complex formation with both human and mouse ASGR1CRD and ASGR2CRD. Compared with the antigens or 8M24-Fab, an early eluting peak (9.10 min), corresponding to the antigen–Fab complex was

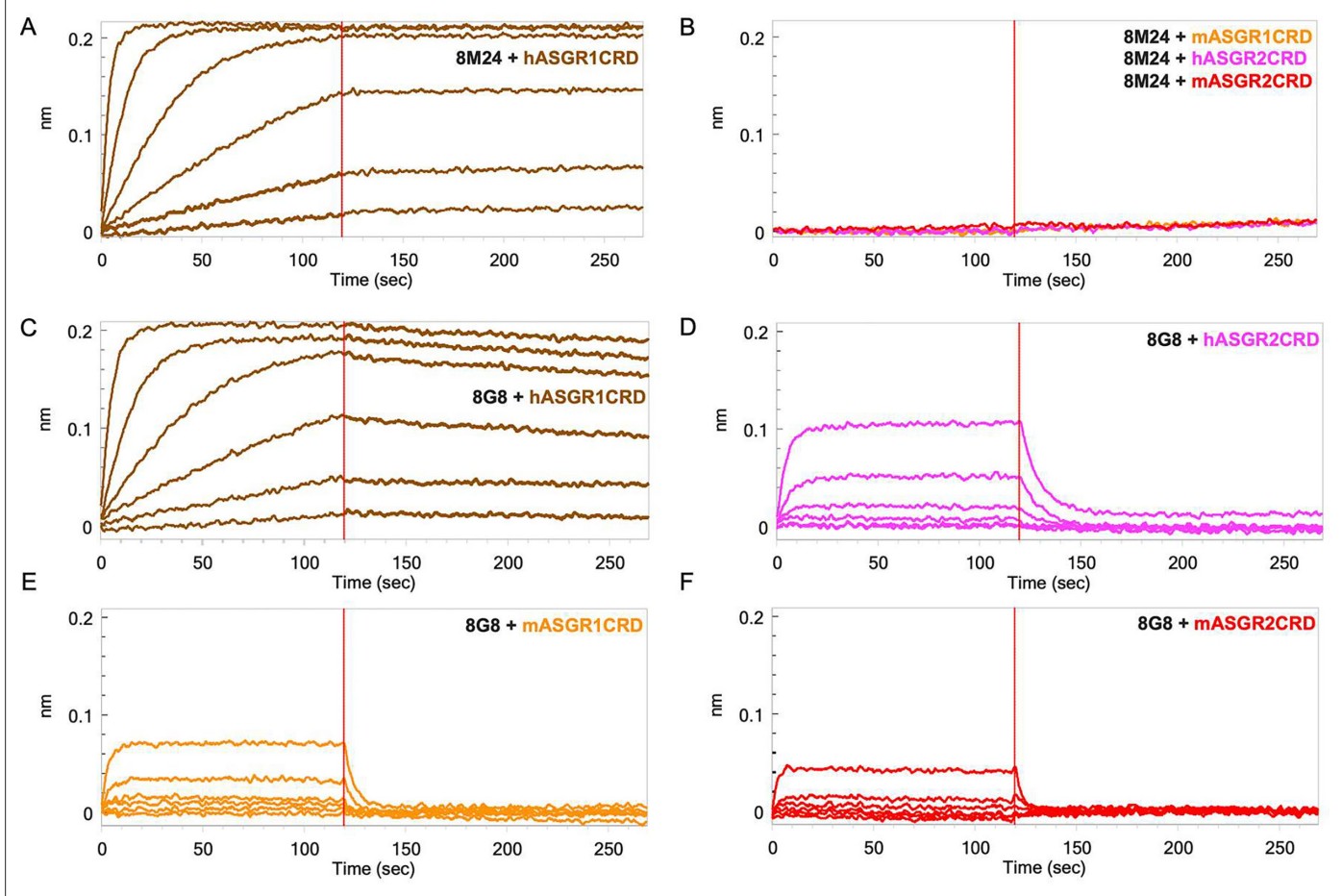

**Figure 1.** Biolayer interferometry (BLI) profiles of antigen carbohydrate recognition domains (CRDs) to antibodies. (**A**) Binding of hASGR1 to 8M24-IgG1; (**B**) non-binding of mASGR1, hASGR2, and mASGR2 to 8M24-IgG1; (**C**) binding of hASGR1 to 8G8-IgG1; (**D**) binding of hASGR2 to 8G8-IgG1; (**E**) binding of mASGR1 to 8G8-IgG1; and (**F**) binding of mASGR2 to 8G8-IgG1. Binding profiles for hASGR1, hASGR2, mASGR1, and mASGR2 are shown in brown, magenta, orange, and red traces, respectively.

The online version of this article includes the following source data and figure supplement(s) for figure 1:

**Source data 1.** Excel file contains raw binding data in *Figure 1*.

**Figure supplement 1.** 8M24 is specific to human ASGR1, and 8G8 binds to both ASGR1 and ASGR2 with mouse cross-reactivity.

**Figure supplement 1—source data 1.** Source data files include raw size-exclusion chromatography (SEC)–high performance liquid chromatography (HPLC) data in *Figure 1—figure supplement 1*.

observed between 8M24-Fab and hASGR1CRD, but not between 8M24-Fab and mASGR1, hASGR2, or mASGR2 CRDs (*Figure 1—figure supplement 1A*). In contrast, early eluting peaks at 9.03, 9.50, 9.35, and 9.50 min were observed for 8G8-Fab with all four antigens (*Figure 1—figure supplement 1B*). These observations suggest that 8M24 binds specifically to hASGR1, while 8G8 is dual specific and binds both ASGR1 and ASGR2.

Furthermore, biolayer interferometry (BLI) was performed to corroborate the SEC results. For BLI, 8M24 and 8G8 IgG1 antibodies were captured on an anti-Human IgG Fc capture (AHC) sensor, and the CRDs of the antigens were used as analytes to determine monovalent affinity. As shown in *Figure 1A*, 8M24 antibody shows tight binding and specific interaction to hASGR1 with $K_D$ in the sub-picomolar range (*Table 1*) and does not bind to mASGR1, hASGR2, and mASGR2 (*Figure 1B*). In contrast, 8G8 antibody binds to all four antigens (*Figure 1C–F*) with the interaction being strongest for hASGR1, followed by those for hASGR2 and mASGR1, and weakest for mASGR2 (*Table 1*). Overall, the BLI results agree with those of SEC, establishing that 8M24 antibody is specific to human ASGR1, while 8G8 binds to both human ASGR1 and ASGR2 and is cross-reactive to their mouse counterparts.

**Table 1.** Kinetic parameters of ASGR1 and ASGR2 binding to 8M24 and 8G8 antibodies.

| Load sample | Analyte | $K_D$ (nM) | $K_D$ (M) | $k_{on}$ (1/Ms) | $k_{dis}$ (1/s) |
|---|---|---|---|---|---|
| 8M24-IgG | hASGR1CRD | <1.0E−3 | <1.0E−12 | 7.39E+05 | <1.0E−07 |
| 8G8-IgG | hASGR1CRD | 1.69 | 1.69E−09 | 5.48E+05 | 9.25E−04 |
| 8G8-IgG | mASGR1CRD | 340 | 3.40E−07 | 5.34E+05 | 1.81E−01 |
| 8G8-IgG | hASGR2CRD | 331 | 3.31E−07 | 2.59E+05 | 8.56E−02 |
| 8G8-IgG | mASGR2CRD | 51,500 | 5.01E−05 | 6.91E+03 | 3.46E−01 |

Note: 8M24-IgG does not bind to mASGR1, hASGR2, and mASGR2 (**Figure 1B**).

Subsequently, we pursued crystal structure determination of hASGR1:8M24 and hASGR2:8G8 complexes to gain molecular insights into their specificities.

## Structure of the hASGR1CRD:8M24 complex

The hASGR1CRD:8M24 complex crystallizes in the $P2_12_12_1$ space group, with one complex molecule per asymmetric unit. The refined structure of hASGR1CRD:8M24 complex, determined at a resolution of 1.7 Å, shows good stereochemistry with $R_{cryst}$ and $R_{free}$ factors of 17.3% and 20.5%, respectively (**Table 2**). The overall structure of hASGR1CRD:8M24 is shown in **Figure 2A**. The structure of ASGR1CRD from its 8M24 complex can be superimposed onto apo-hASGR1CRD (PDB code: 1DV8; **Meier et al., 2000**) with a root mean square deviation (r.m.s.d.) of 0.44 Å over 128 $C^\alpha$ atoms, revealing that the binding of 8M24 to hASGR1 does not induce any significant conformational changes of the antigen. The binding site of 8M24 on hASGR1 is distant from the substrate-binding site marked by glycerol, used as a cryoprotectant while freezing crystals, bound to the $Ca^{2+}$ ion (**Figure 2A, B**). Upon complex formation, both hASGR1 and 8M24 bury an average surface area of 792 Å², and epitope residues of 8M24 on hASGR1 that are within 4.5 Å from the antibody form a close-knit cluster (**Figure 2B**).

The strong interaction between hASGR1 and 8M24, which has sub-picomolar affinity, is predominantly contributed to by residues from the helix ∝1 and those flanking ∝1 (**Figure 2C**), with residues from the β6 strand determining the specificity of 8M24 for hASGR1 over ASGR2 (**Figure 3A**). Lys173 of hASGR1, near the N-terminus of the helix ∝1, forms a hydrogen bond with the main-chain carbonyl oxygen of Thr105 from the HCDR3 loop of 8M24 (**Figure 2D**). Lys173 also interacts strongly with Asp177 within the helix ∝1 of hASGR1, positioning later for hydrogen bonding interaction with Asn32 from the LCDR1 of 8M24. Furthermore, this Asn32 (LCDR1) also interacts with the sidechain of Ser106 from HCDR3. hASGR1 Lys173 is conserved in hASGR2 and substituted with Arg and Leu in mASGR1 and mASGR2, respectively (**Figure 3A**). Introduction of the longer-polar residue Arg or hydrophobic Leu would likely disrupt the structural conformation of closely juxtaposed interacting residues near the N-terminus of the ∝1 helix (**Figure 2D**) and thus might explain lack of interaction between 8M24 and mASGR1 and mASGR2.

Toward the C-terminus of hASGR1 ∝1 helix, conserved or similar residues Leu184, Glu185, and Asp186 are involved in strong hydrogen bonding interactions with 8M24 (**Figures 2E and 3A**). Conserved Glu185 forms ionic and hydrogen bonds with Arg101 and His107 from the HCDR3 of 8M24, and hASGR1 Asp186 forms similar interactions with Arg54 and Asn57 of HCDR2 (**Figure 2E**).

Detailed structural analyses reveal that, in addition to the above mentioned Lys173, Asn180 may also contribute to the specificity of 8M24 for human ASGR1 (**Figure 3B**). The sidechain amide-nitrogen Asn180 of hASGR1 forms a strong hydrogen bond with the mainchain peptide carbonyl-oxygen of Phe91 from the LCDR3 of 8M24. Asn180 also forms a C–H=O hydrogen bond with the $C^\alpha$ atom of Trp92 from the LCDR3 of 8M24. Furthermore, the sidechain of Asn180 fits snugly into a cavity formed by the Phe91–Trp92–Gly93 (LCDR3 loop) and Asn32 (LCDR1 loop) of 8M24, as can be seen from the tight overlap of atomic surfaces formed by these residues (**Figure 3B**). hASGR1 Asn180 is replaced by residues with longer sidechains, such as lysine in mASGR1 and hASGR2, and glutamine in mASGR2. It is likely that residues with longer sidechains are not well accommodated at this position given the tight juxtaposition of Asn180 against the Phe91–Trp92–Gly93 (LCDR3 loop) and Asn32 (LCDR1 loop) of 8M24.

**Table 2.** Crystallography structure determination statistics.

| Data collection | hASGR1CRD:8M24 | hASGR2CRD:8G8 |
|---|---|---|
| PDB code | 8TS0 | 8URF |
| Beamline | ALS BCSB 5.0.2 | ALS BCSB 5.0.2 |
| Wavelength (Å) | 0.9999 | 0.9999 |
| Space group | $P2_12_12_1$ | $H32$ |
| Unit-cell dimensions (Å) | $a$ = 38.9, $b$ = 90.3, $c$ = 167.8 | $a$ = $b$ = 102.41, $c$ = 358.98 |
| Matthew's coefficient (Å³/Da) | 2.17 | 2.71 |
| Solvent content (%) | 43.20 | 54.56 |
| Resolution (Å) | 38.91–1.70 (1.73–1.70) | 38.90–1.85 (1.94–1.90) |
| Number of unique reflections | 66,288 (3439) | 57,655 (3681) |
| Completeness (%) | 100 (100) | 100 (100) |
| $CC_{1/2}$ (%) | 99.9 (40.7) | 99.9 (44.0) |
| $I/\sigma(I)$ | 14.7 (0.9) | 16.3 (1.2) |
| $R_{meas}$ | 0.107 (2.923) | 0.109 (3.273) |
| $R_{pim}$ | 0.030 (0.789) | 0.025 (0.718) |
| Multiplicity | 13.0 (13.6) | 19.8 (20.7) |
| | | |
| Refinement | | |
| Resolution (Å) | 38.05–1.70 (1.73–1.70) | 33.54–1.90 (1.97–1.90) |
| Number of unique reflections | 66,188 (2660) | 57,636 (5730) |
| Number of reflections for $R_{free}$ | 3233 (152) | |
| $R_{cryst}$ (%) | 17.3 | 16.6 |
| $R_{free}$ (%) | 20.5 | 20.35 |
| r.m.s.d.'s from ideal values: | | |
| Bond length (Å) | 0.007 | 0.008 |
| Bond angles (°) | 0.870 | 0.940 |
| Average $B$-factors (Å²): | | |
| Protein | 32.8 | 48.6 |
| Water molecules | 41.0 | 51.6 |
| Ligands | 47.7 | 57.3 |
| Ramachandran plot: | | |
| MolProbity residues in | | |
| Favored region (%) | 97.63 | 98.15 |
| Allowed region (%) | 2.37 | 0.41 |
| Outliers (%) | 0.00 | 0.00 |

Other critical sequence differences in ASGR1 and ASGR2 are noted near the C-terminus of their CRDs (*Figure 3A*). The Thr279 and Leu281 residues, conserved in ASGR1, are replaced in ASGR2 by residues with longer, positively charged sidechains, such as arginine and lysine. The sidechain hydroxyl of hASGR1 Thr279 forms a hydrogen bond with the Arg54 from the HCDR2 of 8M24 (*Figure 2E*). It is likely that the positively charged residues of hASGR2 and mASGR2 near the C-terminus of their CRDs

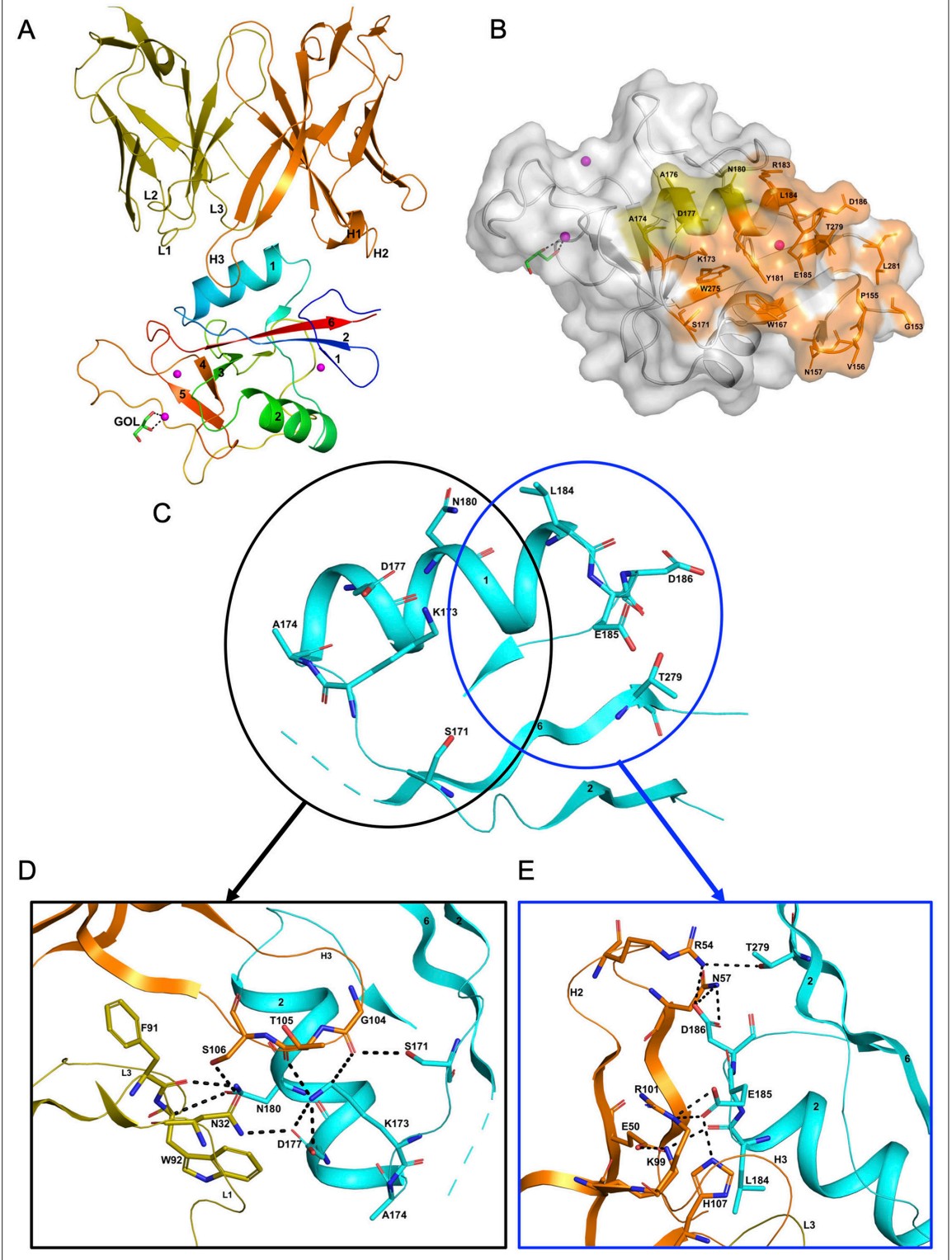

**Figure 2.** Structure of the hASGR1:8M24 complex. (**A**) A cartoon representation of the overall structure of the hASGR1CRD:8M24-Fab complex. The heavy and light chains of 8M24 are colored orange and olive, respectively, with their CDR loops marked as H1, H2, H3, L1, L2, and L3. hASGR1CRD is traced in blue to red from the N- to C-terminus. Three calcium ions are represented as magenta spheres. A glycerol molecule bound to the calcium ion at the ASGR substrate-binding site is highlighted in the stick representation. (**B**) The epitope of 8M24 highlighted on the surface of hASGR1. Antigen residues that are within 4.5 Å from 8M24 heavy and light chains are shown in orange and olive, respectively. (**C**) Residues of hASGR1 involved in polar interactions with the 8M24 antibody are shown in stick notation on the secondary structure elements (cyan traces) with most of the residues situated on

*Figure 2 continued on next page*

*Figure 2 continued*

the helix ∝1 of hASGR1. (**D**) A close-up view of polar interactions between 8M24 and hASGR1 near the N-terminus of helix ∝1. (**E**) A close-up view of polar interactions between 8M24 and hASGR1 near the C-terminus of helix ∝1.

are detrimental for their interaction with 8M24 and may thus contribute toward the specificity of 8M24 for the human ASGR1.

## Structure of the hASGR2CRD:8G8 complex

The hASGR2CRD:8G8 complex crystallizes in the *H*32 space group, with one complex molecule per asymmetric unit. The structure of the hASGR2CRD:8G8 complex was determined by the molecular replacement method at a resolution of 1.9 Å, and refined to $R_{cryst}$ and $R_{free}$ factors of 16.6% and 20.4%, respectively, with good stereochemistry (*Table 2*). The overall structure of hASGR2CRD:8G8 is shown in *Figure 4A*. The structures of hASGR2CRD (determined here) and apo hASGR1CRD (PDB code: 1DV8; *Meier et al., 2000*) could be superimposed with an r.m.s.d. of 0.74 Å over 125 C∝ atoms and a sequence identify of 65%. Thus, the overall fold of ASGR2CRD – made up of six β-strands flanked by two ∝-helices on either side – and three $Ca^{2+}$ ions, which are important for structural stability, are both conserved between the apo hASGR1 and hASGR2 CRDs. This suggests that the binding of 8G8 to ASGR2 does not induce any significant conformational changes in the antigen. Furthermore, the overall structures of the hASGR1 and hASGR2 CRDs from the respective 8M24 and 8G8 complexes can be superimposed on each other with r.m.s.d. values of 0.85 Å over 126 C∝ atoms. Such a superposition reveals that both the 8G8 and 8M24 binds away from the ASGR2 and ASGR1 substrate-binding sites with non-overlapping interaction surfaces, on opposite sides of the antigens, around the helices ∝2 and ∝1, respectively (*Figure 3C*).

8G8 binds to hASGR2CRD between the β1, β2, and β6 strands and the helix ∝2, away from the substrate-binding site marked by glycerol bound to the $Ca^{2+}$ ion (*Figure 4A, B*). Upon complex formation, both hASGR2 and 8G8 bury an average surface area of 663 $Å^2$, and epitope residues of 8G8 on hASGR2 that are within 4.5 Å from the antibody form a close-knit cluster (*Figure 4B*). Among the 17 epitope residues, 5 out of 6 from the β1 strand, 2 out of 3 from the β2 strand, 3 out of 7 from the ∝2 helix, and the Arg297 from the β6 strand are conserved or similar between human, mouse, and rat ASGR1 and ASGR2 (*Figure 3A*). The hASGR2CRD epitope residues Glu183, Lys222, Gln226, His227, Asn229, and Arg297 are involved in hydrogen bonding/ionic interactions with paratope residues from the LCDR3, HCDR2, and HCDR3 of 8G8 (*Figure 4C*). Among these, Glu183, His227, and Arg297 are conserved in human and mouse ASGR1 and ASGR2 (*Figure 3A*), and they are therefore unlikely to contribute toward wide-ranging affinities with the 8G8 antibody (*Table 1*). Asn229, a glycine in h/mASGR1 and a serine in mASGR2, interacts with the heavy-chain Ser57 only through its main-chain peptide carbonyl oxygen (*Figures 3A and 4E*). The side-chain nitrogen of Gln226 is involved in hydrogen bonding interactions with Gln50 (HCDR2) and Thr94 (LCDR3; *Figure 4E*). Gln226 of ASGR2 is diversely substituted by His, Arg, and Lys, respectively, in hASGR1, mASGR1, and mASGR2 (*Figure 3A*). The low-affinity (*Table 1*) binding to mASGR2 may be explained by interaction with Lys222 of hASGR2, which is conserved in hASRG1 (residue Lys200) but substituted by Asn and Asp residues in mASGR1 and mASGR2, respectively (*Figure 3A*). In addition, Lys222 forms strong ionic and hydrogen bonds with Asp93 of LCDR3 (*Figure 4D*). Therefore, an Asp substitution will likely reduce the affinity of mASGR2:8G8 interaction significantly.

Thus, the high-resolution crystal structures have provided molecular insights into the specificities of 8M24 and 8G8 antibodies as well as revealing that their epitopes on ASGR are non-overlapping and away from the glycoprotein-binding site on the receptor.

## 8M24 and 8G8 RSPO2RA fusions lead to enhanced Wnt signaling

RSPO proteins bind to ZNRF3/RNF43 and LGRs via their Furin Fu1 and Fu2 domains, respectively. We have previously engineered a F105R/F109A mutant of human RSPO2 (RSPO2RA), which shows abolished binding to LGRs (*Zhang et al., 2020*). Fusion of RSPO2RA to the anti-ASGR1 antibody 4F3 (4F3-RSPO2RA, also termed 4F3-SWEETS), which binds to the stalk region of ASGR1 outside of the CRD, leads to hepatocyte-specific RSPO mimetic activity (*Zhang et al., 2020*).

Since the binding sites of 8M24 and 8G8 located on the CRD of ASGR are different from that of 4F3, RSPO2RA fusions of 8M24 and 8G8 were constructed to examine the effectiveness of these

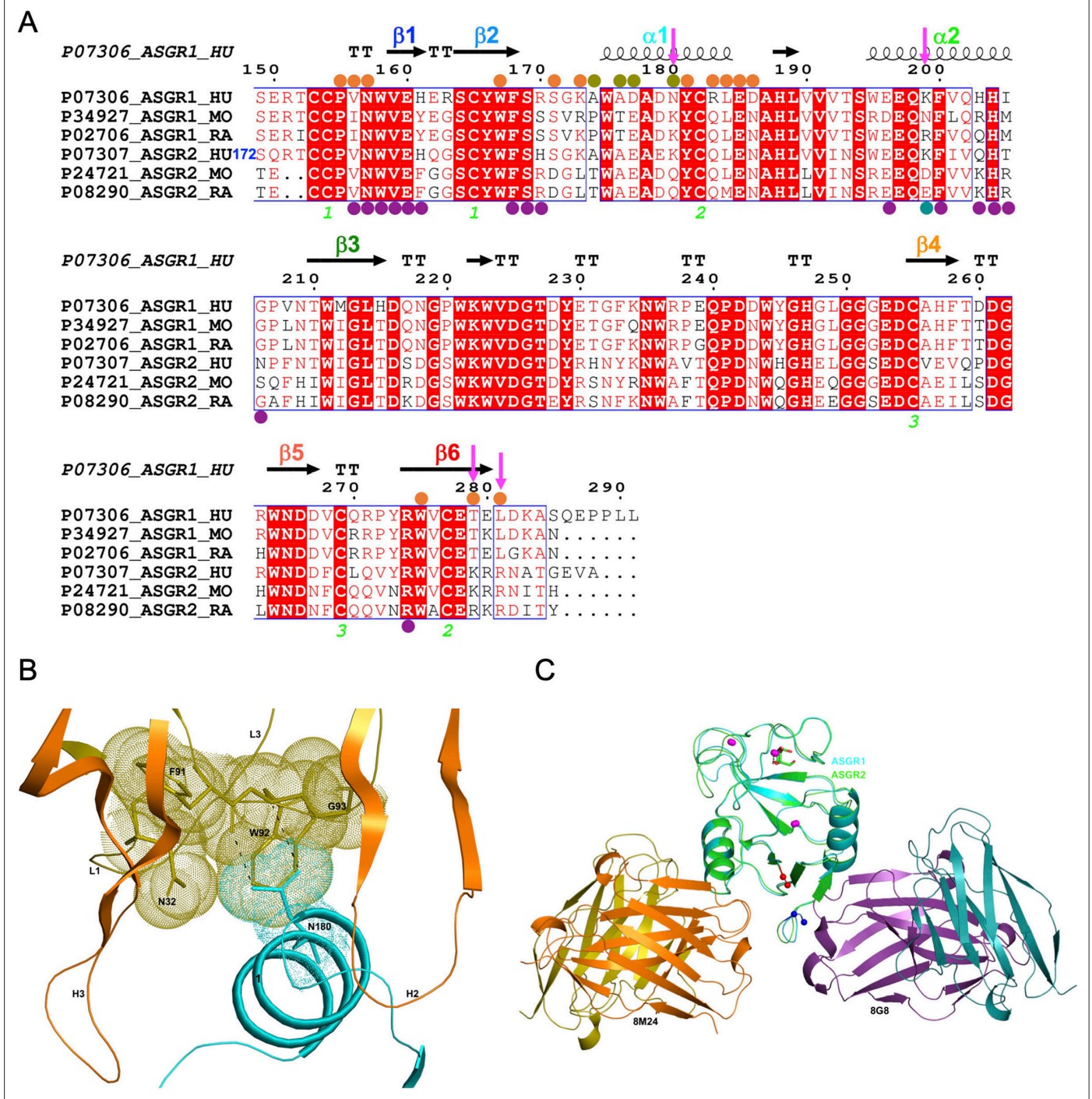

**Figure 3.** Comparison of hASGR1:8M24 and hASGR2:8G8 complex structures. (**A**) An alignment, made with MultAlin (*Corpet, 1988*) and ESPript (*Gouet et al., 2003*) of human, mouse, and rat ASGR1 and ASGR2 sequences with the secondary structure elements of hASGR1CRD (PDB code: 1DV8, *Meier et al., 2000*) depicted at the top and labeled according to the color scheme in *Figure 1A*. Paratope residues from the heavy and light chains of 8M24 are marked with orange and olive circles, respectively. Paratope residues from the heavy and light chains of 8G8 are marked with purple and teal circles, respectively. Important residues are highlighted with a magenta downward arrow. hASGR2 residue Ser172 is marked for reference to the sequence numbering. (**B**) A dot representation showing the snug-fit of hASGR1 Asn180 for the cavity formed by Phe91–Trp92–Gly93 (LCDR3 loop) and Asn32 (LCDR1 loop) of 8M24. hASGR1 Asn180 is replaced by either Lys or Gln in h/m/rASGR2 and m/rASGR1, respectively. (**C**) Superimposed structures of hASGR1:8M24 and hASGR2:8G8 complexes illustrating that the interaction surfaces of the antibodies are situated on the opposite surfaces of the antigens and are non-overlapping. The overall folds of hASGR1 and hASGR2 are shown in cyan and green, respectively, with their N- and C-terminus highlighted as blue and red spheres, respectively. Three calcium ions are represented as magenta spheres. A glycerol molecule bound to the calcium ion at the ASGR substrate-binding site is highlighted in stick representation.

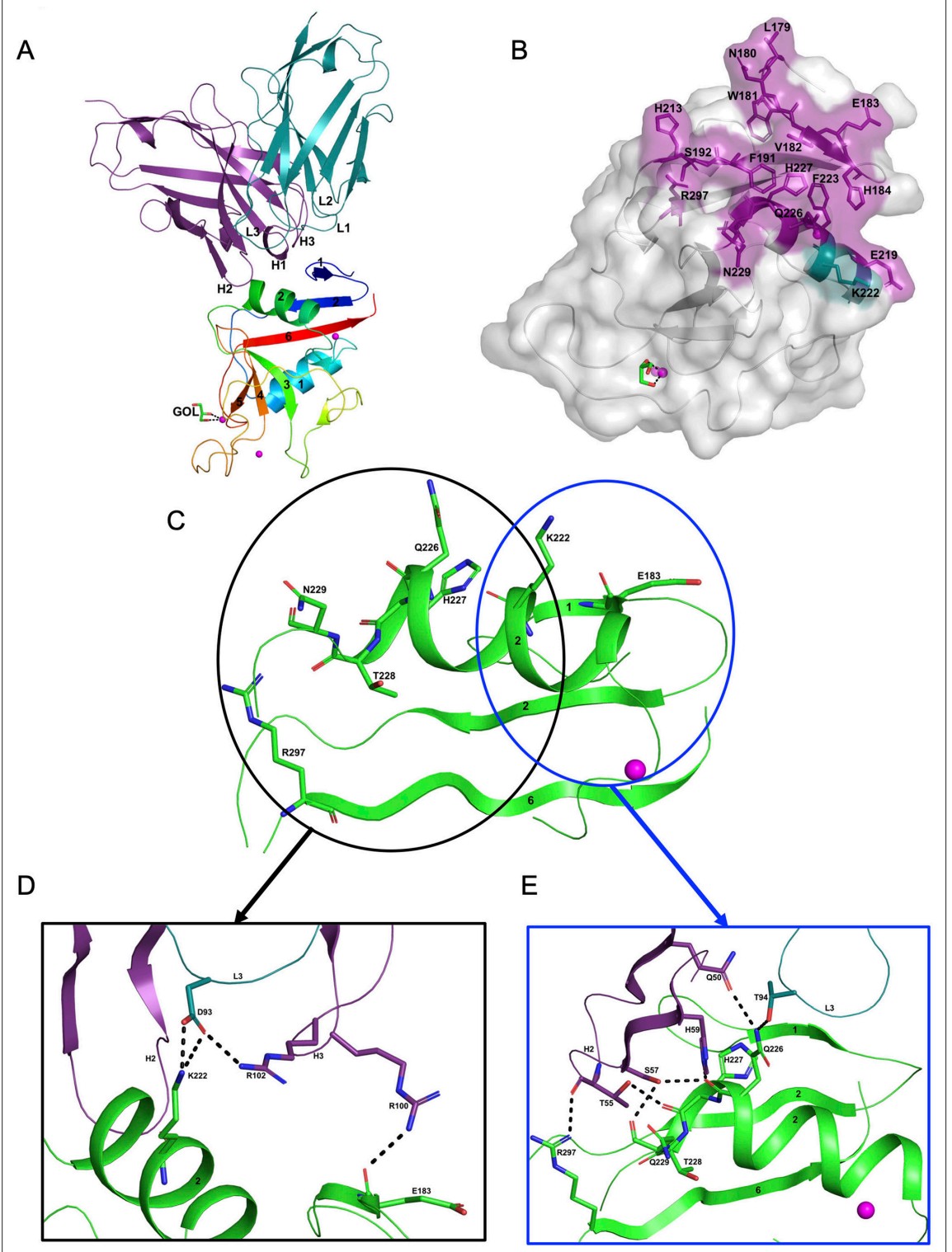

**Figure 4.** Structure of the hASGR2:8G8 complex. (**A**) A cartoon representation of the overall structure of the hASGR2CRD:8G8-Fab complex. The heavy and light chains of 8G8 are colored in purple and teal, respectively, with their CDR loops marked as H1, H2, H3, L1, L2, and L3. hASGR2CRD is traced in blue to red from the N- to C-terminus. Three calcium ions are represented as magenta spheres. A glycerol molecule bound to the calcium ion at the ASGR substrate-binding site is highlighted in stick representation. (**B**) The epitope of 8G8 highlighted on the surface of hASGR2. Antigen residues that are within 4.5 Å from the 8G8 heavy and light chains are shown in purple and teal, respectively. (**C**) Residues of hASGR2 involved in polar interactions with the 8G8 antibody are shown in stick representation on the secondary structure elements (green traces) with most of the residues situated on the

*Figure 4 continued on next page*

*Figure 4 continued*

∝2 helix of hASGR2. (**D**) A close-up view of polar interactions between 8M24 and hASGR2 near the N-terminus of helix ∝2. (**E**) A close-up view of polar interactions between 8M24 and hASGR2 near the C-terminus of helix ∝2.

new ASGR epitopes on RSPO mimetic activity. RSPO2RA is fused to the N-terminus of 8M24 and 8G8 heavy chains using a 15-mer (GGGGS)x3 linker, resulting in 8M24-RSPO2RA and 8G8-RSPO2RA (*Figure 5A*). These two new 8M24- and 8G8-SWEETS molecules also showed robust enhancement of Wnt signaling activity in the presence of a synthetic Wnt source, comparable to that of 4F3-RSPO2RA SWEETS in human hepatocyte HuH-7 cells (*Figure 5B*).

To demonstrate that the RSPO mimetic activity depends on ASGR expression, 8M24-RSPO2RA and 8G8-RSPO2RA were then tested in HEK293 cells, which do not express ASGR. As shown in *Figure 5C*, parental HEK293 cells without ASGR1 overexpression, transfected with an empty negative control vector, respond weakly to the different protein treatments, and no Wnt signaling enhancement is observed by 4F3-RSPO2RA, 8M24-RSPO2RA, or 8G8-RSPO2RA above the negative control

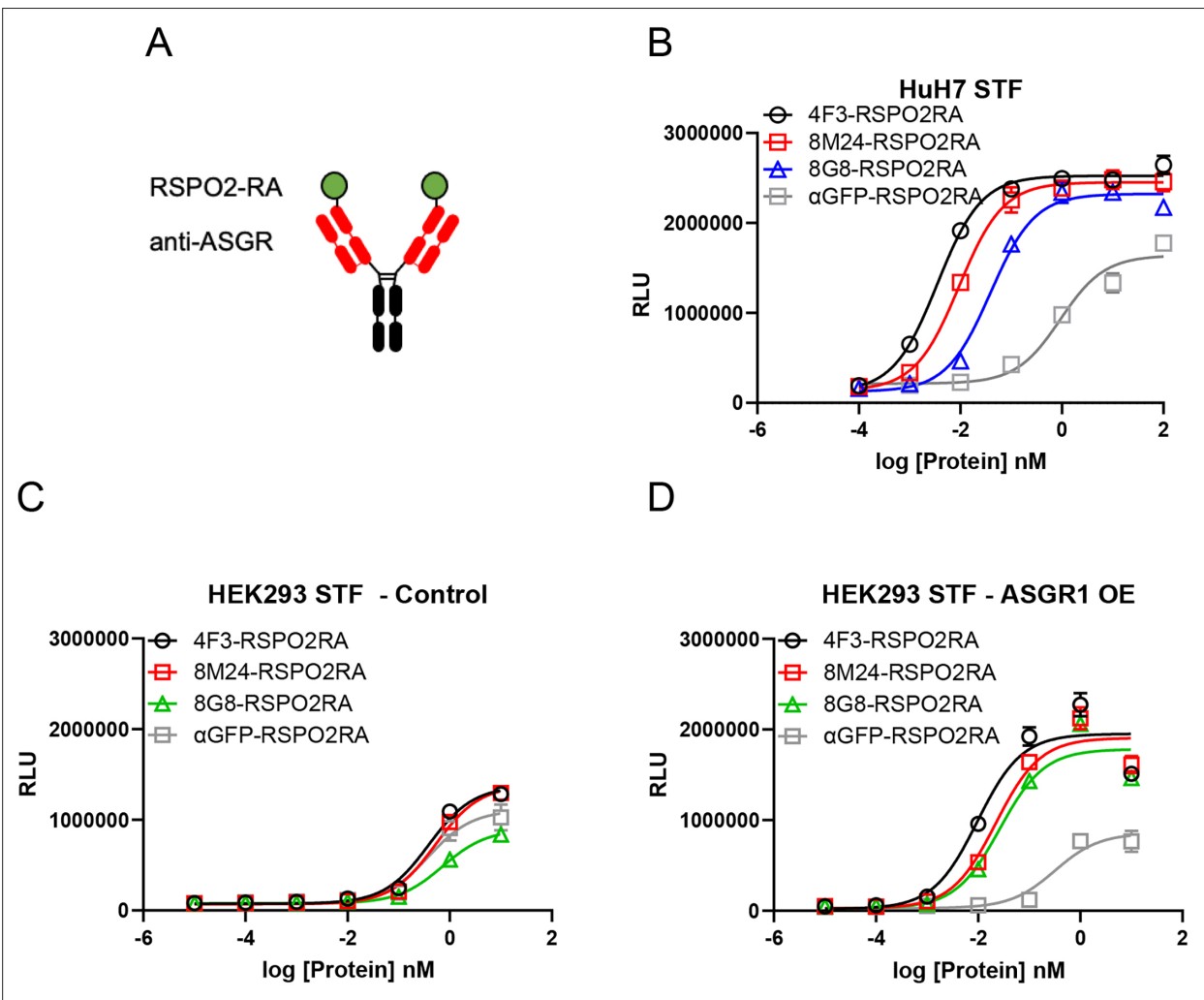

**Figure 5.** Both 8M24 and 8G8 RSPO2RA SWEETS molecules enhance Wnt signaling. (**A**) Diagrams of the SWEETS molecules. RSPO2RA is fused at the N-terminus of the heavy chain of IgG. (**B**) Both 8M24 and 8G8 RSPO2RA SWEETS molecules enhance Wnt signaling in HuH-7 STF cells, which has the Wnt response reporter. (**C, D**) Compared to the negative control αGFP-RSPO2RA, both 8M24 and 8G8 RSPO2RA SWEETS molecules enhance Wnt signaling in ASGR1-overexpressed HEK293 STF cells (**D**), but not in parental HEK293 cells without ASGR1 overexpression (**C**). Data are representative of three independent experiments performed in triplicate and are shown as mean ± standard deviation (SD).

The online version of this article includes the following source data for figure 5:

**Source data 1.** Excel file contains STF raw reading in *Figure 5B–D*.

protein, αGFP-RSPO2RA. In contrast, when HEK293 cells transfected with the ASGR1 expression vector, all three ASGR1-targeted SWEETS molecules show much higher potency than αGFP-SWEETS in enhancing Wnt signaling (*Figure 5D*). These results revealed that the two new epitopes on the CRD of the ASGR receptor, bound by 8M24 and 8G8, also serve as effective epitopes to facilitate clearance of E3 ligase from the cell surface, leading to enhanced Wnt signaling.

## 4F3-RSPO2RA, 8M24-RSPO2RA, and 8G8-RSPO2RA also induce degradation of ASGR1

While our initial goal was to develop liver-specific RSPO mimetic SWEETS molecules to enhance Wnt signaling by ASGR-mediated elimination of E3 ligases, we also wanted to understand whether these bispecific SWEETS molecules have any effect on ASGR1 protein levels. In particular, we wondered if juxtaposing E3 ligases to ASGR by SWEETS binding may induce ubiquitination and subsequent degradation of ASGR proteins as an unintended consequence. Whole-cell extracts from HuH-7 and another hepatocyte cell line, HepG2, treated with various concentrations of the three SWEETS molecules were subjected to western blot analysis with a commercially available anti-ASGR1 antibody (*Figure 6A* and *Figure 6—figure supplement 1A*). The total amount of ASGR1 was reduced in a dose dependent manner after treatment with SWEETS molecules, with 4F3-RSPO2RA and 8M24- RSPO2RA being more potent than 8G8-RSPO2RA. ASGR1 degradation induced by SWEETS molecules was very effective, with only ~20% of the western blot signal remaining at higher concentrations (>1 nM) compared with the untreated samples (*Figure 6A* and *Figure 6—figure supplement 1A*).

Next, we evaluated the kinetics of the SWEETS-mediated ASGR1 degradation. As shown in *Figure 6B* and *Figure 6—figure supplement 1B*, the onset of the degradation is fairly rapid, with ~40% reduction being observed just 4 hr post-treatment (for 4F3- and 8M24-SWEETS), and maximal reduction achieved by 8 hr. Treatments with the anti-ASGR1 antibodies lacking the RSPO2RA domain did not result in the same level of ASGR1 reduction, suggesting that the SWEETS-induced proximity of E3 ligase to ASGR1 indeed facilitates ASGR1 degradation (*Figure 6C* and *Figure 6— figure supplement 1C*).

To determine the route of ASGR1 degradation, we assessed the effect of the autophagy–lysosomal pathway inhibitor bafilomycin A1 and the proteasomal pathway inhibitor MG132 on ASGR1 degradation promoted by SWEETS. As shown in *Figure 6D* and *Figure 6—figure supplement 1D*, either bafilomycin A1 or MG132 treatment significantly impairs ASGR1 degradation, suggesting that SWEETS promotes ASGR1 degradation through both lysosomal and proteasomal pathways. Since these two main degradation pathways have been shown to be linked (because blocking one activates the other) (*Ding et al., 2007*; *Korolchuk et al., 2009*), in addition to assessing the ASGR1 degradation, we examined whole protein ubiquitination and LC3B lipidation to monitor whether each pathway was compromised by inhibitors. Only the MG132-treated group shows a significant increase in whole protein ubiquitination, whereas only the bafilomycin-A1-treated group shows accumulation of autophagosome marker LC3B-II, demonstrating that the inhibitors have little effect on the other degradation pathway (*Figure 6D* and *Figure 6—figure supplement 1D*). Moreover, treatment with the ubiquitin-activating enzyme inhibitor (E1 ligase inhibitor) TAK-243 results in a reduction of ASGR1 degradation, implying that the catalytic function of E3 ligases is involved in SWEETS-mediated ASGR1 degradation (*Figure 6D* and *Figure 6—figure supplement 1D*).

To determine whether ASGR1 is indeed ubiquitinated with SWEETS treatment, we next examined the ubiquitination of ASGR1 by ubiquitin immunoprecipitation (IP). To determine the best timepoint to illustrate ASGR1 ubiquitination, a time-course ubiquitination assay was conducted following 4F3-R-SPO2RA incubation for 1, 2, 4, and 6 hr. As shown in *Figure 7A*, 4F3-RSPO2RA treatment results in enhanced ASGR1 ubiquitination in HepG2 cells, and ASGR1 ubiquitination increases with treatment time, reaching its maximum at 2 hr.

Next, to examine if ASGR1 is ubiquitinated by all three SWEETS treatments, we incubated HepG2 cells with 4F3-, 8M24- or 8G8-RSPO2RA for 2 hr. As shown in *Figure 7B*, all three SWEETS treatments promote ASGR1 ubiquitination over the negative control groups. To further demonstrate that ubiquitination contributes to ASGR1 degradation, we created an ASGR1 mutant (mut ASGR1) in which all the lysine residues in the cytoplasmic domain of ASGR1 are changed to arginine (K3R and K23R) to inhibit potential ubiquitination of ASGR1. Comparing the degradation of mut ASGR1 and wild-type ASGR1 following SWEETS treatment, mutating lysine residues in the cytoplasmic domain of ASGR1

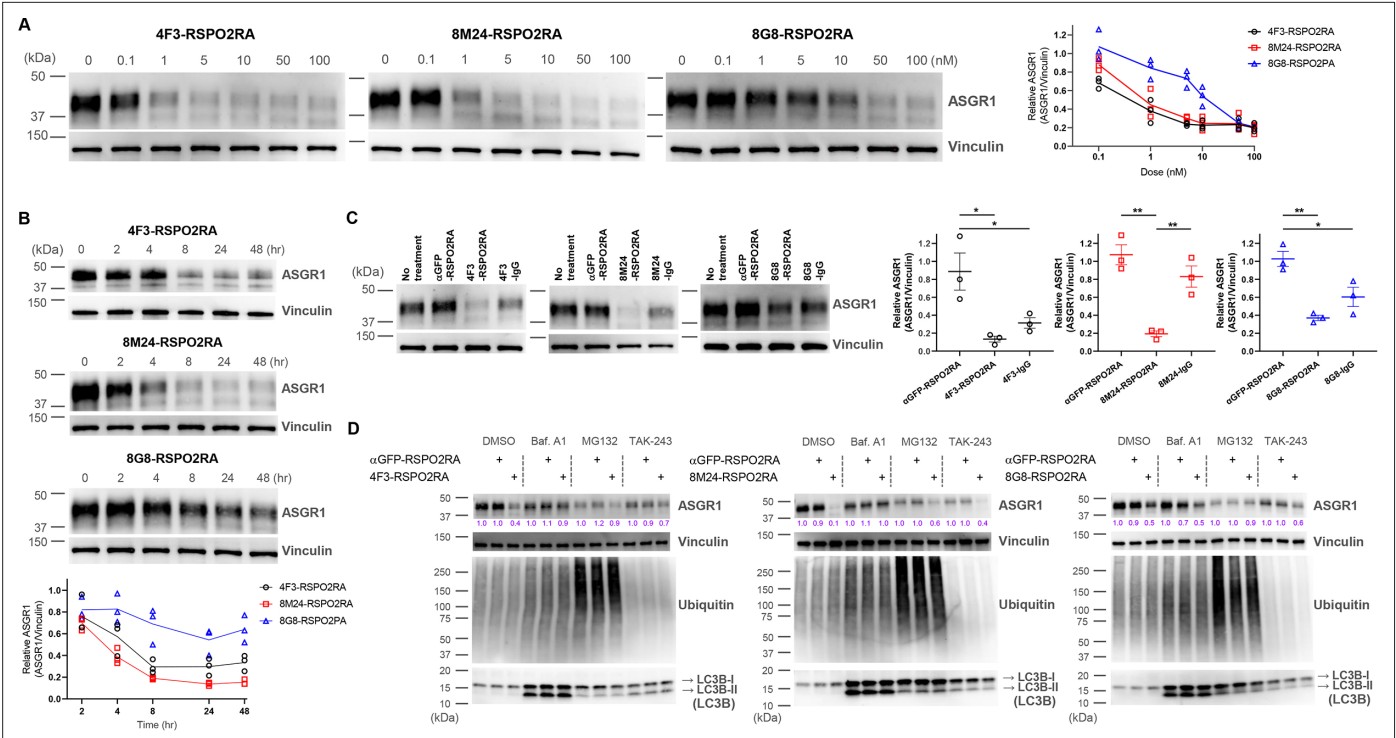

**Figure 6.** SWEETS induce degradation of ASGR1 in HuH-7 cells. (**A**) Dose-dependent ASGR1 degradation promoted by different concentrations of 4F3-, 8M24-, and 8G8-RSPO2RA SWEETS for 24 hr. (**B**) Time-course of ASGR1 degradation upon treatment with 10 nM 4F3-, 8M24-, and 8G8-RSPO2RA. (**C**) Western blot analysis demonstrating the efficacy of ASGR1 degradation in cells treated with 4F3-, 8M24-, and 8G8-RSPO2RA SWEETS compared with ASGR1 antibodies lacking the RSPO2RA domain. (**D**) Western blot data showing the total protein levels of ASGR1, ubiquitin, and LC3B in HuH-7 cells pre-treated with dimethyl sulfoxide (DMSO), lysosomal pathway inhibitor bafilomycin A1 (Baf.A1), proteasome inhibitor MG132, and E1 ubiquitin ligase inhibitor TAK-243 to determine which degradation pathways govern ASGR1 degradation by SWEETS. Data in (**A–C**) are representative of three independent experiments, while data in (**D**) are representative of two independent experiments. For (**A, B**), total ASGR1 levels were normalized to generate graphs representing the mean of those three experiments. In (**C**), data are represented as mean ± standard error of the mean (SEM) of normalized total ASGR1 levels, and one-way analysis of variance (ANOVA) with Tukey's post hoc test was used for statistical analysis. *p < 0.05, **p < 0.01.

The online version of this article includes the following source data and figure supplement(s) for figure 6:

**Source data 1.** Source data files include raw unedited and uncropped blots with the relevant bands clearly labeled shown in *Figure 6A*.

**Source data 2.** Source data files include raw unedited and uncropped blots with the relevant bands clearly labeled shown in *Figure 6B*.

**Source data 3.** Source data files include raw unedited and uncropped blots with the relevant bands clearly labeled shown in *Figure 6C*.

**Source data 4.** Source data files include raw unedited and uncropped blots with the relevant bands clearly labeled shown in *Figure 6D*.

**Source data 5.** Excel file contains the quantification of relative ASGR1 levels in *Figure 6A* right panel, *Figure 6B* bottom panel, and *Figure 6C* right panel.

**Figure supplement 1.** ASGR1 degradation promoted by SWEETS was determined in an additional hepatocyte cell line, HepG2 cells.

**Figure supplement 1—source data 1.** Source data files include raw unedited and uncropped blots with the relevant bands clearly labeled shown in *Figure 6—figure supplement 1A*.

**Figure supplement 1—source data 2.** Source data files include raw unedited and uncropped blots with the relevant bands clearly labeled shown in *Figure 6—figure supplement 1B*.

**Figure supplement 1—source data 3.** Source data files include raw unedited and uncropped blots with the relevant bands clearly labeled shown in *Figure 6—figure supplement 1C*.

**Figure supplement 1—source data 4.** Source data files include raw unedited and uncropped blots with the relevant bands clearly labeled shown in *Figure 6—figure supplement 1D*.

**Figure supplement 1—source data 5.** Excel file contains the quantification of relative ASGR1 levels in *Figure 6—figure supplement 1A* right panel, *Figure 6—figure supplement 1B* bottom panel, and *Figure 6—figure supplement 1C* right panel.

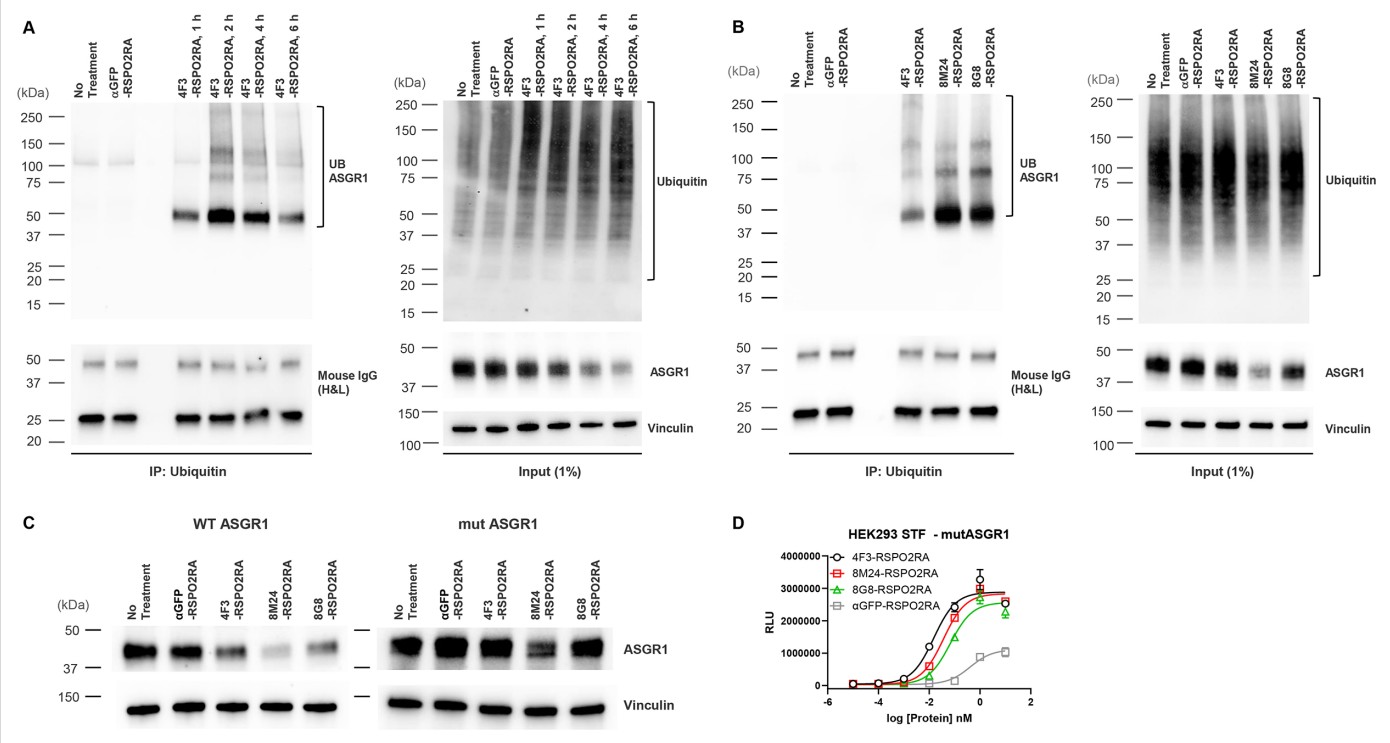

**Figure 7.** SWEETS induce degradation of ASGR1 by recruitment of E3 ubiquitin ligase activity. (**A**) Levels of total and immunoprecipitated ubiquitin and ASGR1 in HepG2 cells subjected to ubiquitin immunoprecipitation (IP) following treatment with 10 nM 4F3-RSPO2RA for the indicated time (1, 2, 4, and 6 hr). The controls (fresh media and 10 nM αGFP-RSPO2RA) were treated for 2 hr before harvest. (**B**) Levels of total and immunoprecipitated ubiquitin and ASGR1 in HepG2 cells subjected to ubiquitin IP following treatment with 10 nM 4F3-, 8M24-, 8G8-RSPO2RA or the controls (fresh media or 10 nM αGFP-RSPO2RA) for 2 hr. (**C**) Western blot analysis results demonstrating ASGR1 degradation in HEK293 STF cells transfected with wild-type ASGR1 and treated with SWEETS compared with cells transfected with mutant ASGR1 that lacks lysine in the cytoplasmic domain. The western blot data are representative of two independent experiments. (**D**) Mutating lysine residues in the cytoplasmic domain of ASGR1 does not affect ASGR1-dependent SWEETS activity. Data are representative of three independent experiments performed in triplicate and are shown as mean ± standard deviation (SD).

The online version of this article includes the following source data for figure 7:

**Source data 1.** Source data files include raw unedited and uncropped blots with the relevant bands clearly labeled shown in *Figure 7A*.

**Source data 2.** Source data files include raw unedited and uncropped blots with the relevant bands clearly labeled shown in *Figure 7B*.

**Source data 3.** Source data files include raw unedited and uncropped blots with the relevant bands clearly labeled shown in *Figure 7C*.

**Source data 4.** Excel file contains STF raw reading in *Figure 7D*.

significantly impair the degradation of ASGR1 (*Figure 7C*). Importantly, mutating lysine residues in the cytoplasmic domain of ASGR1 does not disrupt E3 ligase clearance and Wnt signaling enhancement activity of SWEETS molecules (*Figure 7D*).

Collectively, our data demonstrate that all three SWEETS molecules can act as a TPD platform to degrade not only E3 ligases but also to degrade ASGR through both lysosomal and proteasomal pathways.

## Discussion

TPD technologies have emerged as a promising therapeutic approach over the past two decades (*Alabi and Crews, 2021*; *Luh et al., 2020*; *Zhao et al., 2022*). Protein homeostasis involves a complex network of myriad biochemical processes that mediate protein synthesis, folding, post-translational modification, transport, and removal of damaged proteins. Proteasomes and lysosomes are central to the removal of damaged proteins. Proteasomes degrade transiently lived and misfolded proteins earmarked by the ubiquitin system (*Damgaard, 2021*), while lysosomes facilitate the degradation of long-lived proteins, protein aggregates, entire organelles, and intracellular parasites via endocytosis, phagocytosis, and autophagy. Targeted degradation of intracellular, extracellular, and cell surface

proteins with unwanted functions is an exciting alternative to finding inhibitors, especially in the case of 'undruggable protein targets'. Since the first proof-of-concept by PROTAC (PROteolysis TArgeting Chimeras; *Sakamoto et al., 2001*), a variety of TPD molecular formats, which mainly target intracellular proteins, have emerged, including molecular glue, double mechanism degraders, dTAGs, Trim-away, TF-PROTAC, dual-PROTAC, AUTAC, ATTEC, and AUTOTAC (*Zhao et al., 2022*). Targeted protein degraders that focus on cell surface and extracellular proteins through various mechanisms have also emerged in recent years. These TPD systems can be broadly divided into either lysosomal targeted LYTAC (Lysosomal Targeting Chimera), TransTAC (Transferrin Receptor Targeting Chimera), KineTAC (Cytokine receptor-Targeting Chimera) involving an ASGR or other lysosome shuttling cell surface receptor binder fused to a binder against a protein of interest (POI) (*Ahn et al., 2021*), or mainly proteasomal targeted AbTAC (Antibody-based PROTAC), PROTAB (Proteolysis Targeting Antibody), REULR (Receptor Elimination by E3 Ubiquitin Ligase Recruitment), bispecific aptamer chimera, GlueTAC (Covalent nanobody-based PROTAC), and ROTAC (R-spondin Chimera) (*Zhao et al., 2022*; *Sun et al., 2023*), which are bispecific antibodies that bring E3 ligases to the proximity of a POI.

Our previously described tissue-targeted RSPO mimetic system (SWEETS) is a TPD technology that is applied to Wnt signaling modulation (WO2018/140821, WO2020/014271, *Zhang et al., 2020*). The action of RSPOs is not cell specific due to the broad distribution of their co-receptors which hinders their application in regenerative medicine. By redirecting the E3-binding domain (RSPO2RA) to ASGR1 through a bispecific SWEETS molecule, RNF43 and ZNRF3 can be eliminated specifically from hepatocyte cell surface via ASGR1-mediated endocytosis and lysosomal degradation. Although SWEETS molecules are tissue-targeted RSPO mimetics, this approach of degrading cell surface E3 ligases is similar to TPD systems described as LYTAC (WO2020/132100, *Ahn et al., 2021*) or other lysosome targeting systems.

The previously described ASGR-targeted SWEETS molecule contains an anti-ASGR1 antibody whose epitope is situated on the stalk domain of the receptor (*Zhang et al., 2020*). Here, we have presented structural characterizations of 8M24 and 8G8, two antibodies bound to the CRD of ASGR1 and ASGR2, respectively, distinct from 4F3. These new ASGR epitopes, in the context of 8M24-RSPO2RA and 8G8-RSPO2RA fusion molecules, also confer RSPO mimetic activities similar to those of the stalk-targeted 4F3 antibody. Biophysical experiments revealed that while the 8M24 antibody is specific to human ASGR1, the 8G8 antibody binds to both human ASGR1 and ASGR2 and is also cross-reactive with mouse ASGRs (*Figure 1—figure supplement 1*, *Figure 1*, and *Table 1*). High-resolution crystal structures of hASGR1:8M24 and hASGR2:8G8 Fab complexes revealed (1) the potential structural basis of their specificities toward ASGRs (*Figure 2*, *Figure 3*, and *Figure 4*); (2) that the epitope residues for these two antibodies are situated on opposite surfaces centered around helix ∝1 and helix ∝2, respectively (*Figure 3C*); and (3) neither directly overlaps with the natural ligand pocket (*Figures 2–4*). Furthermore, the structures of hASGR1 and hASGR2 described here also reveal binding sites for glycerol molecules (PDB codes: 8TS0 and 8URF) that can potentially be exploited in designing novel small-molecule binders of ASGR that can be used as ASGR inhibitors, hepatocyte targeters, and/or ASGR-engaging TPDs (*Huang et al., 2017*; *Sanhueza et al., 2017*).

In addition to the LYTAC function-mediated degradation of E3 ligases, we also detected enhanced degradation of ASGR1 by SWEETS, which occurs as a result of bringing E3 ligases (RNF43 and ZNRF3) into the proximity of ASGR1 and subsequent degradation of ASGR1 through the action of proteasomes and lysosomes (*Figure 6*). Therefore, from the perspective of enhanced ASGR1 degradation, the SWEETS molecules also have what are now described as AbTAC (*Cotton et al., 2021*), PROTAB (*Marei et al., 2022*), REULR (*Siepe et al., 2023*), and ROTAC (*Sun et al., 2023*) TPD activities. This E3-mediated degradation is an efficient and rapid process. Despite being one of the most expressed receptors, with an estimate of ~1 million copies in a human hepatocyte (*Miki et al., 2001*), up to ~80% of the ASGR1 was degraded within 8 hr of incubation with SWEETS (*Figure 6*). A rare loss-of-function ASGR1 variant has been associated with a reduced risk of coronary artery disease (CAD) (*Nioi et al., 2016*). Achieving sustained inhibition of ASGR1 via a conventional anti-ASGR1 antibody approach as a potential therapy for CAD may be challenging due to the chronic nature of the disease and high ASGR1 expression levels in hepatocytes. While ASGR elimination was not the initial intended goal for our liver-specific RSPO mimetics, the ability of SWEETS to achieve efficient and sustained degradation of ASGR1 offers a new approach to eliminate this receptor, potentially in the setting of CAD. These results suggest that rapid and efficient degradation of other receptors when linked to

E3 ligases could also occur via this strategy. Given its dual effects on E3 and ASGR1, SWEETS may also combine liver regenerative therapy with treatment for CAD in patients where multiple comorbidities may be present. This dual effect on E3 and its intended target in other TPD systems has also been observed. For example, treatment of an IGF1R-targeting PROTAB (ZNRF3-IGF1R) resulted in not only the reduction of its intended target IGF1R, but also stabilized Wnt receptors, FZD and LRP, and enhanced Wnt signaling as an unintended side effect, similar to SWEETS (*Marei et al., 2022*). While in certain settings, concomitant ASGR1 degradation offers protection against CAD, in other cases, a more specific SWEETS/PROTAB/AbTAC/REULR TPD system might be necessary. Additional design parameters, such as linker length, orientation, epitope, selected E3, stoichiometry of target vs. E3, may need to be optimized to achieve specificity or monoTAC activity.

Finally, beyond LYTAC and AbTAC/PROTAB/REULR functions, SWEETS are ultimately a protein stabilization platform on FZD/LRP receptors, leading to enhanced cellular sensitivity to Wnts. While different from DUBTAC-induced target stabilization (*Henning et al., 2022*), the elimination of E3 ligases specific for the target of interest represents an additional approach for protein stabilization. Therefore, it is interesting to note that ASGR-coupled SWEETS molecules embody multiple 'TAC' activities.

In conclusion, we have described a TPD system that achieves cell-specific enhancement of Wnt signaling. Multiple mechanisms of protein degradation and stabilization are observed with this system. Such an approach could enable a combination of different pathways where beneficial, but our findings also highlight the need to optimize design rules where monoTAC specificity is required. The tools described here could be applied broadly to other applications, for example, anti-ASGR-anti-growth factor receptor or checkpoint inhibitor combinations or anti-E3-anti-growth factor receptor or checkpoint inhibitor combinations could be derived for various cancer treatments. These and other possibilities will be the focus of future studies.

## Materials and methods

**Key resources table**

| Reagent type (species) or resource | Designation | Source or reference | Identifiers | Additional information |
|---|---|---|---|---|
| Cell line (*Homo sapiens*) | Expi293F cells | Thermo Fisher Scientific | A14527 | |
| Cell line (*H. sapiens*) | HEK293 STF | https://doi.org/10.1371/journal.pone.0009370 | | Cells containing a luciferase gene controlled by a WNT-responsive promoter |
| Cell line (*H. sapiens*) | Huh7 STF | https://doi.org/10.1038/s41598-020-70912-3 | | Cells containing a luciferase gene controlled by a WNT-responsive promoter |
| Cell line (*H. sapiens*) | HuH-7 | JCRB Cell Bank | JCRB0403 | |
| Cell line (*H. sapiens*) | Hep G2 | ATCC | HB-8065 | |
| Antibody | Rabbit polyclonal anti-Human ASGR1 antibody | Thermo Fisher | PA5-80356 | RRID:AB_2787681 1:1000 |
| Antibody | Rabbit polyclonal anti-LC3B antibody | Novus | NB100-2220 | RRID:AB_10003146 1:1000 |
| Antibody | Mouse monoclonal anti-Ubiquitin (eBioP4D1) antibody | Thermo Fisher | 14-6078-82 | RRID:AB_837154 1:500 |
| Antibody | Mouse monoclonal anti-Human Vinculin (V284) antibody | Bio-Rad | MCA465GA | RRID:AB_2214389 1:1000 |
| Antibody | Goat polyclonal anti-rabbit IgG H&L (HRP) | Abcam | ab205718 | RRID:AB_2819160 1:20,000 |
| Antibody | Goat polyclonal anti-mouse IgG H&L (HRP) | Abcam | ab205719 | RRID:AB_2755049 1:20,000 |

*Continued on next page*

*Continued*

| Reagent type (species) or resource | Designation | Source or reference | Identifiers | Additional information |
|---|---|---|---|---|
| Peptide, recombinant protein | Fc-R-spondin 2 | https://doi.org/10.1038/s41598-020-70912-3 | | Produced in Expi293F cells |
| Commercial assay or kit | Luciferase Assay System | Promega | E1501 | |
| Commercial assay or kit | Mem-PER Plus Membrane Protein Extraction Kit | Thermo Fisher | 89842 | |
| Commercial assay or kit | Pierce Classic Magnetic IP/Co-IP Kit | Thermo Fisher | 88804 | |
| Commercial assay or kit | Rapid Gold BCA Protein Assay Kit | Thermo Fisher | A53225/A53226 | Discontinued at Thermo Fisher, Use alternatives. |
| Recombinant DNA reagent | ASGR1 | GenScript | OHU03658D | Accession No. NM_001671 |
| Recombinant DNA reagent | pAmCyan1-N1 | Takara Bio | 632442 | |
| Chemical compound, drug | IWP2 | Tocris Bioscience | 3533 | Porcupine inhibitor |
| Software, algorithm | Octet Data Analysis 9.0 | Sartorius | https://www.sartorius.com/en/products/biolayer-interferometry/octet-systems-software | RRID:SCR_023267 |
| Software, algorithm | Prism | GraphPad | https://www.graphpad.com/scientific-software/prism/ | RRID:SCR_002798 |
| Software, algorithm | MOE | Chemical Computing Group | https://www.chemcomp.com/ | RRID:SCR_014882 |
| Software, algorithm | PyMol | Schrödinger | https://pymol.org/ | RRID:SCR_000305 |
| Other | cOmplete His-Tag purification resin | Sigma-Aldrich | 5893801001 | Used for protein purification (methods) |
| Other | CaptivA Protein A affinity resin | Repligen | CA-PRI-0100 | Used for protein purification (methods) |
| Other | Superdex 200 Increase 10/300 GL | Cytiva | 28990944 | Used for protein purification (methods) |
| Other | Anti-hIgG Fc Capture (AHC) biosensors | Sartorius | 18-5060 | Used for protein-binding determination (methods) |

## Cell lines

Expi293F cells were grown in Expi293 Expression Medium (Thermo Fisher Scientific) at 37°C with humidified atmosphere of 8% $CO_2$ in air on an orbital shaker. HEK293 STF cells were maintained in Dulbecco's modified Eagle's medium (DMEM) supplemented with 10% fetal bovine serum (Fisher Scientific) at 37°C in a 5% $CO_2$ environment. Both HuH-7 STF and HuH-7 cells were maintained in DMEM supplemented with 10% fetal bovine serum and non-essential amino acids (Fisher Scientific) at 37°C in a 5% $CO_2$ environment. HepG2 cells were maintained in Eagle's minimum essential medium (EMEM) supplemented with 10% fetal bovine serum at 37°C in a 5% $CO_2$ environment. Cell line authentications were performed at American Type Culture Collection (ATCC) by STR profiling, mycoplasma testing by PCR Mycoplasma detection kit.

## Protein expression and purification

All proteins were cloned into pcDNA3.1(+) vector with the human kappa light-chain secretion signal sequence at the N-terminus and transiently expressed in Expi293 cells using the FectoPro transfection agent (Polyplus Transfection, NY, USA). In 4F3-RSPO2RA, 8M24-RSPO2RA, and 8G8-RSPO2RA SWEETS molecules, the F105R/F109A mutant of human RSPO2 furin domain was fused at the N-terminus of the heavy chain of the respective antibodies via a 15-mer (G4S) × 3 linker. hASGR1, mASGR1, hASGR2, mASGR2, 8M24-Fab, and 8G8-Fab were expressed with His-tag and purified using cOmplete His-Tag purification resin (Roche, USA) following standard protocols. 4F3, αGFP, 8M24, and

8G8 antibodies and 4F3-RSPO2RA, 8M24-RSPO2RA, and 8G8-RSPO2RA SWEETS were purified using either CaptivA resin (Repligen, USA) under gravity-flow or using a prepacked MabSelect SuRe (Cytiva, USA) column attached to an AKTA FPLC. All proteins were further polished on a Superdex 200 size-exclusion chromatography column equilibrated with 2× HBS (HEPES-Buffered Saline) (40 mM HEPES pH 7.5, 300 mM sodium chloride) buffer.

## Determination of binding affinity and specificity using BLI

For the BLI assays, the 8M24 and 8G8 antibodies were prepared at 50 nM in 1× phosphate buffer saline + (0.05%) (PBST) + 0.5 mg/ml bovine serum albumin (assay buffer) (Teknova P1192, Fisher BP1600) and captured on Anti-hIgG Fc Capture (AHC) biosensors (Sartorius 18-5060). Capture was recorded for 40 s. Association was measured by transferring the IgG-loaded sensors to wells containing CRDs of hASGR1, mASGR1, hASGR2, and mASGR2 at a threefold dilution series from 1 µM to 4 nM for 120 s. Dissociation was measured by placing sensors in wells containing only assay buffer for 150 s. Assays were performed in duplicate. Kinetic parameters were determined using instrument analysis software to fit a global 1:1 kinetic model with Rmax linked across all concentrations.

## hASGR1CRD:8M24 and hASGR2CRD:8G8 crystallization and structure determination

Purified hASGR1CRD and hASGR2CRD were mixed with the 8M24 and 8G8 Fabs, respectively, at 1.1:1 molar ratio and incubated with carboxy-peptidase A and B at a wt/wt ratio of 100:1 overnight at 4°C. Complex formation was confirmed by observation of a single major peak on a SuperdexS200 Increase (10/300 GL) column pre-equilibrated in 20 mM HEPES pH 7.5 and 150 mM sodium chloride, and by sodium dodecyl sulfate–polyacrylamide gel electrophoresis. Initial crystallization screening (hASGR-1CRD:8M24 at 20 mg/ml and hASGR2CRD:8G8 at 28 mg/ml) using commercially available reagents was performed using Mosquito (TTP LabTech) liquid handler and equilibrated at 18°C inside an Echo-Therm incubator (Torrey Pines Scientific, USA). Crystallization conditions were further optimized by grid-screens or microseed matrix screens (*D'Arcy et al., 2014*). Diffraction-quality crystals of the hASGR1CRD:8M24 complex grew in buffer containing 100 mM SPG (succinic acid, sodium phosphate monobasic monohydrate, glycine buffer) at pH 9.0 and 25% wt/vol PEG 1500. Crystals were cryo-protected using 20% glycerol in the well solution. Diffraction-quality crystals of the hASGR2CRD:8G8 complex grew in buffer containing 100 mM sodium HEPES at pH 7.5, 100 mM calcium chloride, and 30% (wt/vol) PEG 400. Crystals were cryo-protected using 16% glycerol in the well solution. X-ray diffraction datasets were collected at the Berkeley Center for Structural Biology at the Advanced Light Source (ALS), Berkeley, CA, and processed with XDS (*Kabsch, 2010*) and xdsme (*Legrand, 2017*) programs. Structures were determined by the molecular replacement method using Phaser (*McCoy et al., 2007*) with a poly-alanine model of ASGR1CRD (PDB code: 5JPV; *Sanhueza et al., 2017*) and variable and constant domains Fab, followed by refinement using Phenix (*Liebschner et al., 2019*) and validation by MolProbity (*Williams et al., 2018*). Crystallography models were manually inspected and built using COOT. Analyses of refined crystal structures and image creation were performed using MOE (CCG, https://www.chemcomp.com/) and PyMol (https://pymol.org/).

## SuperTop Flash assay

Wnt-signaling activity was measured using cell lines containing a luciferase gene controlled by a Wnt-responsive promoter (STF reporter) as previously reported (*Janda et al., 2017*). In brief, cells were seeded in 96-well plates at a density of 10,000 per well 24 hr prior to treatment, then treated with various proteins overnight in the presence of 100 pM Wnt mimetic FA-L6 (*Fowler et al., 2021*). Cells were lysed with Luciferase Cell Culture Lysis Reagent (Promega, Madison, WI) and activity was measured with Luciferase Assay System (Promega, Madison, WI) using vendor-suggested procedures. Data were plotted as average −/ + standard deviation of triplicate and fitted by non-linear regression using Prism (GraphPad Software, San Diego, CA). For overexpression of exogenous receptors, cells were transiently transfected with plasmids containing receptors of interest under eukaryotic expression promoters (ASGR1 was clone OHU03658D from GenScript, negative control pAMCyan1 from Takara), then split into 96-well plates (20,000 cells per well) for SuperTop Flash (STF) assay 24 hr post-transfection.

## Western blot analysis

For the ASGR1 degradation assay, HuH-7 and HepG2 cells at a density of $1 \times 10^5$ cells per well (24-well plate) were plated a day before the experiment, then treated with 10 nM 4F3-RSPO2RA, 8M24-RSPO2RA, 8G8-RSPO2RA, 4F3-IgG, 8M24-IgG, 8G8-IgG, or the control (αGFP-RSPO2RA) for 24 hr. For the dose–response experiments, indicated concentrations of SWEETS or controls were treated for 24 hr. For the time-course experiments, each SWEETS molecule (10 nM) was incubated for the indicated duration. For the ASGR1 degradation comparison assay between wild-type ASGR1 transfection and mutant ASGR1 transfection, HEK293 STF cells at a density of $1 \times 10^6$ cells per well (6-well plate) were plated a day before the experiment. The cells were transfected with vectors containing either wild-type ASGR1 or mutant ASGR1 using the Lipofectamine 3000 Transfection Reagent (Invitrogen, L3000008). Cells were split into five wells (12-well plate) 30 hr after transfection and subjected to different treatment conditions (10 nM 4F3-RSPO2RA, 8M24-RSPO2RA, 8G8-RSPO2RA, or the controls, i.e., fresh cell media or 10 nM αGFP-RSPO2RA) for 24 hr. When lysosomal pathway inhibitor bafilomycin A1 (10 nM, Millipore Sigma 1661), proteasome inhibitor MG132 (1 µM, Selleck Chemicals S2619), E1 ubiquitin ligase inhibitor TAK-243 (50 nM, Selleck Chemicals S8341), or DMSO (as the control vehicle) was treated, inhibitors were pre-treated for 18 hr, 10 nM of SWEETS or control (αGFP-RSPO2RA) was then introduced for 6 hr in the presence of inhibitors or DMSO. Following incubation, cells were washed in ice-cold TBST and lysed in RIPA (Radioimmunoprecipitation Assay Buffer (25 mM Tris-HCl, pH 7.6, 150 mM NaCl, 1% NP-40, 1% sodium deoxycholate, 0.1% SDS)) buffer (Thermo Fisher, 89900) supplemented with 1% Halt protease inhibitor (Thermo Fisher, 87786), 1% Halt phosphatase inhibitor (Thermo Fisher, 78420), and 0.1% Benzonase (Millipore Sigma, E1014) on ice for 20 min. Whole-cell lysates were prepared by collecting supernatants after centrifugation at $14,000 \times g$ for 20 min at 4°C. Protein concentration was determined by BCA assay (Thermo Fisher, A53225/A53226). Samples were prepared by mixing with 4× Laemmli Sample Buffer (Bio-Rad, 1610747) supplemented with BME (2-mercaptoethanol) and incubated for 5 min at 95°C. Samples prepared with equal amounts of proteins were run on 10% Mini-PROTEAN TGX Precast Gels (Bio-Rad, 4561033) (for ASGR1 blot) or 4–20% Mini-PROTEAN TGX Precast Gels (Bio-Rad, 4568093) (for Ubiquitin and LC3B blot) using Tris/Glycine/SDS running buffer (Bio-Rad, 1610732). Gels were transferred to nitrocellulose membranes (Bio-Rad, 1704158EDU) using the Trans-Blot Turbo Transfer System (Bio-Rad). The membranes were blocked with SuperBlock T20 (TBS) Blocking Buffer (Thermo Fisher, 37536) for 1 hr at room temperature. Following blocking, the membranes were incubated with primary antibodies overnight at 4°C. Blots were washed three times with TBST for 5 min each and subsequently incubated with horseradish peroxidase (HRP) secondary antibodies for 1 hr at room temperature. Blots were then washed three times with TBST for 5 min each, developed using enhanced chemiluminescent (ECL) HRP substrates (Pierce, 34580), and visualized using ChemiDoc Imaging System (Bio-Rad). Image Lab (Bio-Rad) was used to quantify the signal intensities of the bands.

## ASGR1 ubiquitination assay

ASGR1 ubiquitination with antibody treatment was evaluated in HepG2 cells. Cells ($3 \times 10^6$) were plated in 10 cm dishes a day before the experiment. For ubiquitination time-course experiments, cells were treated with 10 nM 4F3-RSPO2RA or the controls (fresh media or 10 nM αGFP-RSPO2RA) at 37°C for the indicated duration. For ASGR1 ubiquitination experiments, cells were treated with 10 nM 4F3-RSPO2RA, 8M24-RSPO2RA, 8G8-RSPO2RA, or the controls (fresh media or 10 nM αGFP-RSPO2RA) at 37°C for 2 hr. Following treatment, membrane proteins were extracted using the appropriate kit (Thermo Fisher, 89842) following the manufacturer's instructions with adjustment. In brief, cells were scraped and resuspended in growth media, then harvested by centrifugation at $300 \times g$ for 5 min. Cell pellets were washed twice in 1.5 and 0.75 ml of Cell Wash Solution and collected by centrifugation at $300 \times g$ for 5 min. Cytosolic proteins were extracted by incubating the cell pellet with 0.75 ml Permeabilization Buffer for 10 min at 4°C with constant mixing. The supernatant containing cytosolic proteins was removed from the permeabilized cells after centrifuging for 15 min at $16,000 \times g$. Membrane proteins were extracted by incubating the cell pellet with 0.6 ml Solubilization Buffer for 1 hr at 4°C with constant mixing. The mixture was centrifuged at $16,000 \times g$ for 15 min at 4°C, and the membrane-associated proteins were transferred to a new tube followed by protein concentration determination by BCA assay (Thermo Fisher, A53225/A53226). For ubiquitin IP at 3.4 mg per sample, IP/Co-IP kit (Thermo Fisher, 88804) was used, and the membrane protein lysates were

incubated with 10 µg mouse monoclonal anti-ubiquitin (eBioP4D1) antibody (Invitrogen, 14-6078-82) at 4°C overnight. The immune complex samples were incubated with 25 µl prewashed Pierce Protein A/G Magnetic Beads per sample for 1 hr at room temperature. The beads were washed twice with 500 µl IP Lysis/Wash Buffer and once with 500 µl of ultra-pure water following incubation, and proteins were eluted with 28 µl of Laemmli Sample Buffer (Bio-Rad, 1610747). For IP antibody control, 1 µl of each elution sample was supplemented with BME and prepared for western blot analysis. For input samples, 0.34 µg were used from the prepared IP samples prior to IP antibody incubation for western blot analysis.

### Statistical analyses

All statistical analyses were performed using GraphPad Prism 9.0 (GraphPad Software, San Diego, CA, USA). Detailed information about statistical analyses is stated in the figure legends.

## Acknowledgements

The authors would like to thank Haili Zhang, Asmiti Sura, Ralph McAnelly, and Hayoung Go for technical support and Randall Brezski and Sean Bell for helpful discussions. We thank Craig Parker for critical reading of the manuscript and Priya Handa for editorial support. Authors also acknowledge the help of staff at the Berkeley Center for Structural Biology which is supported by the Howard Hughes Medical Institute and the National Institutes of Health, National Institute of General Medical Sciences, ALS-ENABLE grant P30 GM124169. The Advanced Light Source is a Department of Energy Office of Science User Facility under Contract No. DE-AC02-05CH11231.

## Additional information

### Competing interests

Parthasarathy Sampathkumar, Heekyung Jung, Zhengjian Zhang, Zhong Huang, Tom Lopez, Robert Benisch, Wen-Chen Yeh: former full-time employee and shareholder of Surrozen, Inc. Hui Chen, Nicholas Suen, Yiran Yang, Sung-Jin Lee, Jay Ye: current full-time employee and shareholder of Surrozen, Inc. Yang Li: current full-time employee and shareholder of Surrozen, Inc; is Executive Vice President of Research at Surrozen, Inc.

### Funding

| Funder | Grant reference number | Author |
|---|---|---|
| Surrozen Inc | | Parthasarathy Sampathkumar Heekyung Jung Hui Chen Zhengjian Zhang Nicholas Suen Yiran Yang Zhong Huang Tom Lopez Robert Benisch Sung-Jin Lee Jay Ye Wen-Chen Yeh Yang Li |

The funders had no role in study design, data collection, and interpretation, or the decision to submit the work for publication.

### Author contributions

Parthasarathy Sampathkumar, Heekyung Jung, Conceptualization, Resources, Formal analysis, Supervision, Investigation, Visualization, Methodology, Writing – original draft, Writing – review and editing;

Hui Chen, Formal analysis, Supervision, Investigation, Visualization, Methodology, Writing – review and editing; Zhengjian Zhang, Conceptualization, Formal analysis, Supervision, Investigation, Visualization, Methodology, Writing – review and editing; Nicholas Suen, Yiran Yang, Zhong Huang, Tom Lopez, Robert Benisch, Sung-Jin Lee, Jay Ye, Resources, Investigation, Visualization, Methodology, Writing – review and editing; Wen-Chen Yeh, Conceptualization, Resources, Supervision, Funding acquisition, Writing – review and editing; Yang Li, Conceptualization, Resources, Formal analysis, Supervision, Funding acquisition, Visualization, Methodology, Writing – original draft, Writing – review and editing

### Author ORCIDs
Nicholas Suen ⓘ https://orcid.org/0000-0002-6388-024X
Robert Benisch ⓘ https://orcid.org/0000-0002-0612-7606
Yang Li ⓘ https://orcid.org/0000-0002-7134-5685

Reviewer #1 (Public Review): https://doi.org/10.7554/eLife.93908.3.sa1
Author response https://doi.org/10.7554/eLife.93908.3.sa2

## Additional files

### Supplementary files
• MDAR checklist

### Data availability
The atomic coordinates and structure factor of the hASGR1CRD:8M24 and hASGR2CRD:8G8 complex structures are available in the Protein Data Bank with accession codes 8TS0 and 8URF, respectively. All other data associated with this study are included in the paper or the supplementary files.

The following datasets were generated:

| Author(s) | Year | Dataset title | Dataset URL | Database and Identifier |
|---|---|---|---|---|
| Sampathkumar P, Li Y | 2024 | Crystal Structure of human ASGR1 CRD (Carbohydrate Recognition Domain) bound to 8M24 Fab | https://www.rcsb.org/structure/8TS0 | RCSB Protein Data Bank, 8TS0 |
| Sampathkumar P, Li Y | 2024 | Crystal Structure of human ASGR2 CRD (Carbohydrate Recognition Domain) bound to 8G8 Fab | https://www.rcsb.org/structure/8URF | RCSB Protein Data Bank, 8URF |

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
