## [Editor Report · eLife assessment]

The manuscript describes a **valuable** method to boost WNT signaling in a tissue-specific manner. The work extends previous data from the authors based on fusing an RSPO2 mutant protein to an antibody that binds ASGR1/2. In the current manuscript, two new antibodies with similar effects are described, that expand this **solid** approach and provide alternatives for potential future clinical applications. This manuscript will be of interest to all scientists studying protein engineering and cellular targeting.

---

## [Referee Report · Reviewer #1 (Public Review)]

Summary:

The authors have previously described a way to boost WNT/CTNNB1 signaling in a tissue-specific manner, by directing an RSPO2 mutant protein (RSPO2RA) to a liver-specific receptor (ASGR1/2). This is done by fusing the RSPO2RA to an antibody that binds ASGR1/2.

Here the authors describe two new antibodies, 8M24 and 8G8, with similar effects. 8M24 shows specificity for ASGR1, while 8G8 has broader affinity for mouse/human ASGR1/2.

The authors resolve and describe the crystal structure of the hASGR1CRD:8M24 complex and the hASGR2CRD:8G8 complex in great detail, which help explain the specificities of the 8M24 and 8G8 antibodies. Their epitopes are non-overlapping.

Upon fusion of the antibodies to an RSPO2RA (an RSPO mutant), these antibodies are able to enhance WNT signaling by promoting the ASGR1-mediated clearance of ZNRF3/RNF43, thereby increasing cell surface expression of FZD. This has previously also been shown to be the case for RSPO2RA fused to an anti-ASGR1 antibody 4F3 - and the paper also tests how the antibodies compare to the 4F3 fusion.

Strengths:

(1) One challenge in treating diseases, is the fact that one would like therapeutics to be highly specific - not just in terms of their target (e.g. aimed at a specific protein of interest) but also in terms of tissue specificity (i.e. affecting only tissue X but leaving all others unaffected). This study broadens the collection of antibodies that can be used for this purpose and thus expands a potential future clinical toolbox.

(2) The authors have addressed questions raised after a first round of review, e.g. by showing that ASGR1 is itself indeed ubiquitinated.

Weaknesses:

(1) Some questions remain as to how 8M24 and 8G8 compare to 4F3.

(2) Some questions remain as to the specificity of the approach: the initial goal was not to also downregulate ASGR1 per se, so this targeting to a specific receptor/membrane protein is not trivial and/or neutral.

---

## [Author Response]

The following is the authors’ response to the original reviews.

**Public Reviews:**

**Reviewer #1 (Public Review):**
Weaknesses:The authors demonstrate that ASGR1 is degraded in response to RSPO2RA-antibody treatment through both the proteasomal and the lysosomal pathway, suggesting that this is due to the RSPO2RA-mediated recruitment of ZNRF3/RNF43, which have E3 ubiquitin ligase activity. The paper doesn't show, however, if ASGR1 is indeed ubiquitinated.

We thank the reviewer for this comment. We have now conducted ASGR1 ubiquitination assays by immunoprecipitation (IP) of ubiquitin in the membrane protein extract, and immunoblotting (IB) ASGR1 after treating HepG2 cells with our SWEETS molecules or controls. The new data demonstrated ubiquitination of ASGR1 with SWEETS treatment (new Fig. S3A and S3B). Additionally, we blocked the potential ubiquitination of ASGR1 by mutating the two lysine residues in the cytoplasmic domain and compared the ASGR1 degradation after SWEETS treatment. The new data show that removing the potential ubiquitylation Lys sites prevented ASGR1 degradation post SWEETS treatment (new Fig. S3C). These new results provide direct evidence that ASGR1 is ubiquitinated to undergo lysosome or proteasome degradation.

The authors conclude that the RSPO2A-Ab fusions can act as a targeted protein degredation platform, because they can degrade ASGR. While I agree with this statement, I would argue that the goal of these Abs would not be to degrade ASGR per se. The argumentation is a bit confusing here. This holds for both the results and the discussion section: The authors focus on the dual role of their agents, i.e. on promoting both WNT signaling AND on degrading ASGR1. They might want to reconsider how they present their data (e.g. it may be interesting to target ASGR1, but one would presumably then like to do this without also increasing WNT responsiveness?).

We thank the reviewer for this comment. As the reviewer states, the initial goal of the RSPO2RA-ab fusions was to generate tissue-specific RSPO mimetics that focus on elimination of E3. As an unintended consequence, we observed enhanced elimination of ASGR as well. While this was unintended, the results did provide POC that when an E3 ligase is brought into proximity of another protein, ubiquitination and degradation of this protein may occur. Additionally, our results highlight that one needs to be careful in fully assessing the impact of bispecific molecules on the intended target as well as unintended targets to understand the potential side effects of such bispecific molecules. We have revised the manuscript to make this more clear, both in the Results and Discussion sections.

Lines 326-331: The authors use a lot of abbreviations for all of the different protein targeting technologies, but since they are hinting at specific mechanisms, it would be better to actually describe the biological activity of LYTAC versus AbTAC/PROTAB/REULR so non-experts can follow.

We thank the reviewer for this suggestion. We have added more details in the Discussion to highlight the different mechanisms of the various systems described.

Can the authors comment on how 8M24 and 8G8 compare to 4F3? The latter seems a bit more specific (ie. lower background activity in the absence of ASGR1 in 5C)? Are there any differences/advances between 8M24 and 8G8 over 4F3? This remains unclear.

These three antibodies bind different regions/epitopes on ASGR. 8M24 and 8G8 bind non-overlapping epitopes on the carbohydrate recognition domain (CRD), while 4F3 binds the stalk region outside of the CRD. This information is in the Results section of the manuscript. We do not believe that the difference in the ASGR binding epitopes contributes to the slight differences in the background activity. The slight differences may be due to differences in the conformation of the antibodies resulting from the differences in their primary sequences, and these differences may not be significant. We have now repeated the experiments in Fig. 5C and 5D to address the reviewer’s next comment on the axis. These new data (new Fig. 5C and 5D) show less background differences between the molecules.

Can the authors ensure that the axes are labelled/numbered similarly for Fig 5B-D? This will make it easier to compare 5C and 5D.

We thank the reviewer for this suggestion. The y-axes in Fig. 5B–D now have the same scale and number format. For Figs. 5C and 5D, we focus on the potency increases of the SWEETS molecules post ASGR1 overexpression.

**Reviewer #2 (Public Review):**
Weaknesses:The authors show crystal structures for binding of these antibodies to ASGR1/2, and hypothesize about why specificity is mediated through specific residues. They do not test these hypotheses.

We thank the reviewer for this comment. We did not further test the residue contributions to binding and specificity as this is not the main focus of the current manuscript. We have revised the section and tuned down the claims for specificity.

The authors demonstrate in hepatocyte cell lines that these function as mimetics, and that they do not function in HEK cells, which do not express ASGR1. They do not perform an exhaustive screen of all non-hepatocyte cells, nor do they test these molecules in vivo.

We agree with the reviewer. For the 4F3-based SWEETS molecule, additional in vitro and in vivo specificity characterized were performed and described in Zhang et al., Sci Rep, 2020. Since 8M24 is human specific and 8G8 only weakly interacts with mouse receptors, in vivo experiments in mouse were not performed. While we did not extensively test the 8M24- and 8G8-based SWEETS on additional cell lines or in vivo, we do believe the data presented strongly support the hepatocyte-specific effects of these molecules.

Surprisingly, these molecules also induced loss of ASGR1, which the authors hypothesize is due to ubiquitination and degradation, initiated by the E3 ligases recruited to ASGR1. They demonstrate that inhibition of either the proteasome or lysosome abrogates this effect and that it is dependent on E1 ubiquitin ligases. They do not demonstrate direct ubiquitination of ASGR1 by ZNRF3/RNF43.

We thank the reviewer for this comment. We have now conducted ASGR1 ubiquitination assays by immunoprecipitation (IP) of ubiquitin in the membrane protein extract, and immunoblotting (IB) ASGR1 after treating HepG2 cells with our SWEETS molecules or controls. The new data demonstrate ubiquitination of ASGR1 with SWEETS treatment (new Figs. S3A and S3B). Additionally, we blocked the potential ubiquitination of ASGR1 by mutating the two lysine residues in the cytoplasmic domain and compared the ASGR1 degradation after SWEETS treatment. The new data show that removing the potential ubiquitylation Lys sites prevented ASGR1 degradation post SWEETS treatment (new Fig. S3C). These new results provide direct evidence that ASGR1 is ubiquitinated to undergo lysosome or proteasome degradation.

**Recommendations for the authors:**

**Reviewer #1 (Recommendations For The Authors):**
There are multiple instances where articles (i.e. the use of "the") are missing.

We thank the reviewer for this comment. Following the suggestion, the manuscript has gone through a detailed review by an editorial service, and these and other grammatical errors have been corrected.

**Reviewer #2 (Recommendations For The Authors):**
The best I can think of is to inject these into Wnt reporter mice (or maybe humanized mice) and see if the liver lights up while other tissues do not.

We thank the reviewer for this suggestion. The liver specificity was demonstrated in vivo in our earlier publication (SciRep, 10:13951, 2020) with the 4F3-RSPO2RA molecule. Unfortunately, as the results in this manuscript show, the new ASGR binders 8M24 and 8G8 either do not bind or only weakly interact with mouse receptors. Therefore, the in vivo experiments were not performed here.

You could also consider addressing some of the statements in the manuscript that are currently hypothetical experimentally.

We thank the reviewer for this comment. We did not further test the residues’ contribution to binding and specificity as this is not the main focus of the current manuscript. We have revised the section and tuned down the claims for specificity.

It would be easier to compare the graphs in 5B-D if all Y-axes were the same scale, with the same scientific notation.

We thank the reviewer for this suggestion. The y-axes in Fig. 5B-D now have the same scale and number format. For Figs. 5C and 5D, we focus on the potency increases of the SWEETS molecules post ASGR1 overexpression.

Some of the western blots in Figure 6 do not have antibody/target labels, making them harder to interpret.

All the Western blots antibody/target labels are on the right side of the blots for each panel, we have now made the text bold and thus easier to identify.

Figure 6 and Supplementary Figure 2 are the same I think.

Figure 6 and Supplementary Figure 2 show the same experimental set-up performed on two different cell lines, Fig. 6 is on Huh7 cells and Supplementary Fig. 2 is on HepG2 cells. The results from these two cell lines are quite consistent, making their appearance very similar.